# Successful Needle Aspiration of a Traumatic Pneumothorax: A Case Report and Literature Review

**DOI:** 10.3390/medicina60040548

**Published:** 2024-03-28

**Authors:** Giuseppe Bettoni, Silvia Gheda, Michele Altomare, Stefano Piero Bernardo Cioffi, Davide Ferrazzi, Michela Cazzaniga, Luca Bonacchini, Stefania Cimbanassi, Paolo Aseni

**Affiliations:** 1Department of Emergency Medicine, ASST Grande Ospedale Metropolitano Niguarda, 20162 Milan, Italy; giuseppe.bettoni@ospedaleniguarda.it (G.B.); silvia.gheda@ospedaleniguarda.it (S.G.); davide.ferrazzi2@ospedaleniguarda.it (D.F.); michela.cazzaniga@ospedaleniguarda.it (M.C.); luca.bonacchini@ospedaleniguarda.it (L.B.); 2Advanced Technologies in Surgery, Department of Surgical Sciences, University of Rome Sapienza, 00185 Rome, Italy; michele.altomare@ospedaleniguarda.it (M.A.); stefanopiero.cioffi@ospedaleniguarda.it (S.P.B.C.); 3General Surgery Trauma Team, ASST Grande Ospedale Metropolitano Niguarda, 20162 Milan, Italy; stefania.cimbanassi@ospedaleniguarda.it; 4Department of Pathophysiology and Transplantation, University of Milan, 20122 Milan, Italy; 5Department of Biomedical and Clinical Sciences “L. Sacco”, Università degli Studi di Milano, 20157 Milan, Italy

**Keywords:** pneumothorax, needle aspiration, traumatic pneumothorax, chest aspiration, conservative treatment, emergency department

## Abstract

Traumatic pneumothorax (PTX) occurs in up to 50% of patients with severe polytrauma and chest injuries. Patients with a traumatic PTX with clinical signs of tension physiology and hemodynamic instability are typically treated with an urgent decompressive thoracostomy, tube thoracostomy, or needle decompression. There is recent evidence that non-breathless patients with a hemodynamically stable traumatic PTX can be managed conservatively through observation or a percutaneous pigtail catheter. We present here a 52-year-old woman who presented to the emergency department with a 55 mm traumatic PTX. Following aspiration of 1500 mL of air, a clinical improvement was immediately observed, allowing the patient to be discharged shortly thereafter. In hemodynamically stable patients with a post-traumatic PTX, without specific risk factors or oxygen desaturation, observation or simple needle aspiration can be a reasonable approach. Although the recent medical literature supports conservative management of small traumatic PTXs, guidelines are lacking for hemodynamically stable patients with a significantly large PTX. This case report documents our successful experience with needle aspiration in such a setting of large traumatic PTX. We aimed in this article to review the available literature on needle aspiration and conservative treatment of traumatic pneumothorax. A total of 12 studies were selected out of 190 articles on traumatic PTX where conservative treatment and chest tube decompression were compared. Our case report offers a novel contribution by illustrating the successful resolution of a sizable pneumothorax through needle aspiration, suggesting that even a large PTX in a hemodynamically stable patient, without other risk conditions, can be successfully treated conservatively with simple needle aspiration in order to avoid tube thoracostomy complications.

## 1. Introduction

Traditionally, thoracic injury has accounted for approximately 25% of trauma-related mortality, with up to 40–50% of these patients likely to have a pneumothorax (PTX) [1]. PTX is defined as the presence of air in the pleural space and is broadly classified as spontaneous or non-spontaneous secondary to trauma or iatrogenic. The most common clinical practice is to use chest tubes for traumatic pneumothoraces in order to prevent further enlargement, which can result in tension physiology and severe hypotension from obstructive shock. In ventilated trauma patients in the pre-hospital setting with an impending tension PTX, bilateral finger thoracostomy, starting on the side with the suspected tension PTX, is a valuable temporary measure followed by definitive tube thoracostomy when the patient arrives in the trauma center and is stabilized.

In patients with a traumatic pneumothorax, current Advanced Trauma Life Support (ATLS) guidelines recommend chest tube placement, with the caveat that asymptomatic patients with small pneumothoraces who are not ventilated may be managed with observation or aspiration at the discretion of the providing physician. While observation and aspiration are both acceptable options according to ATLS guidelines, they are considered to be at risk of inducing a tension pneumothorax [2]. However, chest drain insertion in a traumatic pneumothorax is not without risk and complications such as malpositioning, malfunction, and empyema are reported with a frequency ranging from 1 to 25%. Some severe and potentially fatal complications such as injuries to the heart, great vessels, spleen, and liver and perforation of the esophagus have also been reported [3,4,5]. According to the majority of studies in which patients were conservatively treated, traumatic pneumothoraces were described as “small”, “minimal”, or “moderate” and chest drains were always used to treat those described as “large”, “complete”, or “total”. Furthermore, patients with chest blunt trauma have also been reported to experience an occult pneumothorax in 7.8% of cases, and in the majority of non-ventilated patients with an occult pneumothorax a conservative approach has been reported [6,7]. Although current data on observation vs. chest tube placement are conflicting and difficult to interpret due to the different patient populations studied [8,9,10], recent studies have suggested that conservative management for patients with traumatic penumothoraces is a possible option for small–moderate traumatic pneuomothoraces, as defined by the 35 mm rule, and patients with a PTX less than 35 mm can be treated by initial observation regardless of the mechanism of injury [11]. Some of these patients can benefit from needle aspiration. Nevertheless, in traumatic PTX, needle aspiration vs. conventional chest tube insertion has never been prospectively evaluated. Only a few prospective randomized controlled trials have been performed comparing conservative and non-conservative management of traumatic pneumothoraces using chest drains. Herein, we present a case of a large pneumothorax in a blunt chest trauma patient that resolved with needle aspiration. We also aimed to conduct a systematic review summarizing the existing evidence on needle aspiration in traumatic pneumothorax as well as the available evidence supporting conservative management in traumatic PTX.

## 2. Case Presentation

Following a car accident, a 52-year-old Italian woman with no significant medical or surgical history presented to the local French Emergency Department (ED). A diagnostic workup with a chest X-ray and CT revealed one non-displaced rib fracture in her right fifth rib, a right-sided pneumothorax, and a fracture of the transverse processes of T11–T12–L1–L2. The patient refused admission despite medical advice to accept hospitalization and returned to Italy, where she immediately presented to our Emergency Department. She complained of pain in her right chest and tenderness on palpation of the dorsal spine (Visual Analog Scale score, 6). A repeated chest X-ray revealed a right-sided pneumothorax measuring 55 mm (Figure 1a). The patient’s blood pressure was 135/86, her heart rate was 71/minute, and she was eupneic with a respiratory rate of 18/min with an oxygen saturation of 96%. We decided to perform needle aspiration after multidisciplinary discussions with our trauma surgeons. A 14 Fr cannula was inserted under local anesthesia in the second intercostal space near the midclavicular line in a sterile field. A total of 1500 mL of air was aspirated. The patient was admitted to our medical unit for close observation. Serial chest X-rays performed on days 1 and 2 demonstrated a significant reduction in pneumothorax size (Figure 1b). The patient was discharged after 4 days with a chest X-ray showing a 5 mm right pneumothorax. Follow-up at 10 days by chest X-ray showed complete resolution of the pneumothorax flap. As a result of our case report, we thought it would be helpful to provide a review of the current evidence related to conservative treatment for traumatic PTX that should not be considered a clinical guidance document. Rather, it provides the available evidence about clinical practice in the current medical literature addressing some of the variation that may exist in the management of traumatic pneumothorax.

## 3. Materials and Methods

### 3.1. Research Methodology

This literature review adheres to the Preferred Reporting Items for Systematic Reviews and Meta-Analysis (PRISMA) guidelines (see Figure 2). 

Preliminary review registration in the PROSPERO database was considered unnecessary due to the scarcity of data due to the limited number of articles available on the subject and the small number of controlled prospective trials making a meta-analysis impossible.

The inclusion criteria were as follows: All studies or articles that specifically address the review’s research question or topic were considered, including randomized controlled trials (RCTs), cohort studies, case-control studies, systematic reviews, meta-analyses, observational studies, and retrospective studies. Only studies that involve the specific intervention of needle aspiration or conservative treatment following traumatic PTX were considered. The search was conducted across three databases, including Metacrawler, PubMed, and EMBASE, from January 1976 to October 2023 by employing the MeSH terms “Pneumothorax” AND “Trauma” AND “Conservative Management”.

The exclusion criteria were as follows: Duplicate publications or multiple reports of the same study, editorials, commentaries, letters, conference abstracts, and other non-peer-reviewed sources, and studies conducted on animals were excluded. We excluded articles published in languages other than English. Additionally, we excluded all articles that did not include all three MeSH terms. Only peer-reviewed articles were considered.

### 3.2. Technical Details of Pneumothorax Needle Aspiration

Ideally, the patient should lie supine with the trunk raised 15 degrees. In supine patients, the air can be suctioned out from the second intercostal space above the rib in order to avoid the neurovascular bundle. An ultrasound check with a linear probe could be useful in order to exclude the presence of anatomical variants of the mammary artery, which usually is located more medially. The skin and subcutaneous tissues are injected with lidocaine using a 10 mL syringe. The air is aspirated when the pleural space is reached. The pleural space is then infiltrated, and the needle is retracted while infiltrating the superficial planes. A 14-gauge cannula is then connected to another 10 mL syringe containing 5 mL of physiological solution (if there are no specific needle aspiration devices) and inserted at 90 degrees. The needle is advanced a few millimeters and then removed, leaving the plastic sheath behind. Being cautious not to kink it, the cannula is connected to the extension cable, the 3-way tap, and the 50 mL Luer-Lock syringe and aspiration is started while counting how many milliliters of air are suctioned (using the 3-way tap to throw the aspirated air into the environment) until resistance is felt. Once this is performed, the cannula is retracted a little bit and aspiration and counting are continued. As soon as there is no more air to suction, the cannula is removed and the dressing is applied. When more than 2.5 L of air (50 syringes) is aspirated, the procedure is considered to have failed [12].

## 4. Results

As a result of our search, we found 190 articles using the three MeSH terms “Pneumothorax” AND “Trauma” AND “Conservative Management”. However, 178 papers were excluded for the following reasons: (1) the articles did not address traumatic pneumothoraces and conservative management simultaneously, (2) the articles were linked to other severe injuries, including vascular, abdominal, and tracheobronchial injuries, (3) the articles referred to penetrating types of chest injuries, and (4) the articles referred to iatrogenic injuries. A meticulous analysis was conducted of 22 relevant publications evaluating conservative treatment for patients with a traumatic PTX. We selected 12 articles [13,14,15,16,17,18,19,20,21,22,23,24] where needle aspiration was also discussed as a conservative treatment option. We summarize all data extracted from these 12 studies in Table 1, which includes the first author, the year of publication, the study reference, patient groups, the study type, and outcomes and key results.

## 5. Discussion 

Although the conservative management and needle aspiration of spontaneous pneumothorax were described 58 years ago [25] in some patients without a hospital admission, a practice originally suggested by the experience with artificial pneumothorax when treating pulmonary tuberculosis, the conservative management of traumatic PTX is a relatively recent treatment option.

Traumatic PTX is commonly seen in severely traumatized patients with blunt chest trauma following motor vehicle collisions. A pneumothorax is diagnosed with a combination of physical exam findings and imaging. Most patients’ primary complaint is shortness of breath due to pain during inspiration that is usually caused by a fractured rib.

Traditionally, traumatic PTXs have been managed with chest tubes, but several studies have questioned whether conservative management can also be utilized in selected patients. The 10th edition of ATLS: Advanced Trauma Life Support states that “any traumatic PTX is best treated with a chest tube” because of the possibility of the development of a tension pneumothorax. 

Although the same guidelines state that a physician can observe occult PTXs, they do not provide precise guidelines regarding size or other parameters [2]. Although effective, chest tubes are invasive procedures associated with increased morbidity, extended hospital stays, and complications including malpositioning, infection, re-expansion pulmonary edema, and many other life-threatening complications such as cardiovascular, spleen, liver, and esophageal injuries.

Although trauma physicians can accurately detect traumatic pneumothoraces with chest X-rays, an extended Focused Assessment with Sonography in Trauma (eFAST) has become the mainstay in the initial assessment of trauma patients. The advent of CT imaging has increased the identification of occult PTXs, defined as a PTX detected on CT but not suspected on clinical evaluation or chest X-rays. This increased sensitivity has led surgeons to question whether tube decompression is really necessary for small PTXs, and some authors have suggested that this CT finding of an occult PTX does not have an impact on patient outcome and, therefore, no treatment can be necessary other than a period of observation [7,14,15].

Recent studies have confirmed that hemodynamically stable patients with small PTXs can be observed. Mahmood et al. [7] examined blunt trauma patients with an occult PTX and, among the 85% that were managed conservatively (no tube thoracostomy), 3.9% eventually required chest tube placement during the course of their hospital care.

In 2019, Eddine BZS and Coll [11] developed the “35 mm rule”, providing evidence that observation can be safe for the management of PTXs < 35 mm on CT scan imaging. This retrospective study looked at 288 patients with blunt and penetrating traumatic PTXs < 35 mm in size. Sizing was performed by measuring the distance between the parietal and visceral pleura in the largest air pocket. In the same study, 257 patients (89.0%) were successfully monitored until discharge. Although the 35 mm rule was limited to patients requiring CT scans, there was general agreement that some sort of objective measurement of the pneumothorax should be used to guide decisions. 

The main question was whether stable patients with traumatic pneumothoraces could be treated safely and effectively without a chest drain. Additionally, it was also necessary to understand which patients would be the most appropriate for conservative treatment.

A retrospective observational study on 602 patients by Walker et al. [18] reviewed the prospectively collected Trauma Audit and Research Network (TARN) database to identify patients with traumatic pneumothoraces. Investigators used multivariable Cox regression analysis to determine which factors were independently predictive of failure of conservative management. Demographic, injury, management, and pneumothorax characteristics were obtained from the database, with pneumothorax size determined by chest radiograph and CT imaging. From the total of 602 patients with traumatic pneumothorax identified in the database, 277 (46%) were initially managed conservatively without needle decompression, chest tube insertion, or chest surgery. Of the 277 patients that were managed conservatively, 252 (90%) did not require subsequent intervention, including the majority (56/62, 90%) of patients requiring positive pressure ventilation (PPV). There was no difference in the risk of failure of conservative treatment between ventilated and non-ventilated patients. Failure of conservative management was not predicted by the initial size of the pneumothorax, the injury severity score (ISS), the presence of rib fractures, or bilateral vs. unilateral pneumothoraces. Hemothorax (>2 cm) alone predicted failure of conservative treatment. In the same study, the authors concluded that, when deemed clinically safe by the treating physician, the majority of conservatively managed patients with traumatic pneumothorax can be successfully treated without the need for a chest drain, regardless of ventilatory status. The authors acknowledged that their study lacked a high percentage of all penetrating chest injuries (5%), possibly affecting generalizability. Due to the retrospective nature of the study and the lack of a control group over the decision to intervene, the implications of this study should be cautiously accepted. 

Based on these two large retrospective studies [11,18], there seems to be a clear shift toward a more conservative approach to the management of traumatic PTX. 

In addition, several RCTs and a Cochrane study [26] have become available in patients with a primary spontaneous PTX supporting aspiration rather than a chest tube or small catheter thoracostomy. These RCTs have shown elevated efficacy rates, shorter hospital stays, and fewer complications in patients with a primary spontaneous pneumothorax using aspiration as initial treatment despite an immediate success rate for chest tube insertion being observed, but with a longer hospital stay. 

Despite this growing consensus on the conservative management of small traumatic PTXs, there are still no specific and clear indications for conservative management in hemodynamically stable patients with significantly traumatic PTXs.

We aimed to contribute in this review to the ongoing discussion about management strategies for a subset of stable patients with large-sized PTXs by emphasizing these distinctive features and limitations of the literature where precise indications for conservative management are still unclear. Our case report describing a successful needle aspiration experience in the context of a large-sized PTX may provide another valuable insight into traumatic conservative therapy in a patient with hemodynamically stable conditions more than 12 h after the car accident. A significant contribution to the decision on conservative management was the multidisciplinary discussion with trauma team surgeons, who also recommended needle aspiration as the first line of treatment. As a matter of fact, needle aspiration offered both diagnostic and therapeutic benefits in this case. By relieving the pressure from the pneumothorax, we aimed to alleviate the patient’s symptoms and prevent further deterioration, thus potentially avoiding the need for more invasive interventions like chest tube insertion. The decision to proceed with needle aspiration involved a careful evaluation of the risks and benefits. Risks associated with any invasive procedure, including infection and pneumothorax exacerbation, were evident; however, the potential benefits of prompt intervention outweighed these risks in the context of our patient’s clinical presentation and stability.

To the best of our knowledge, our case report is the first reported case in the literature of a patient with a large traumatic pneumothorax with complete resolution after needle aspiration.

A few considerations on PTX size can help to highlight that our case is noteworthy for the successful resolution of a large traumatic pneumothorax with an inter-pleural distance measuring 55 mm. There is a substantial difference between this size and the 3.5 cm threshold commonly suggested for conservative management. By demonstrating the efficacy of needle aspiration in such cases, our report challenges existing paradigms. It expands the spectrum of feasible interventions for patients with large traumatic PTXs in selected clinical settings with hemodynamic stability and without other associated risk factors. Another remarkable aspect of our case is the timing of the intervention. Needle aspiration was performed approximately 12 h post-trauma, suggesting its utility even in a delayed setting. This temporal dimension adds another valuable perspective, particularly in scenarios where immediate intervention may not be feasible or indicated.

It is important to consider the possibility of needle aspiration failure in a traumatic pneumothorax, which can involve a number of factors including inadequate placement of the needle as well as an inability to completely evacuate the air from the pleural space due to loculated air pockets. Aspiration failure has been reported more frequently in patients with a traumatic PTX with an inter-pleural distance > 20 mm at the level of the hilum. Additionally, if the pneumothorax is large or under tension, needle aspiration alone may not be sufficient to achieve adequate lung re-expansion [27].

In cases where the initial needle aspiration fails to fully resolve the pneumothorax, it may be necessary to consider a repeat needle aspiration. Some controversies exist on the decision to repeat the procedure in traumatic PTX. The decision strictly depends on the clinical status of the patient, the size of the pneumothorax, and the expertise of the healthcare provider performing the procedure. It is possible to perform a second aspiration in patients with a traumatic PTX in the absence of respiratory compromise if the PTX persists or recurs. However, when more than 2.5 L of air is aspirated, the procedure should be considered to have failed [12]. The time frame from the acute event to the procedure can be an important variable in the success rate of needle aspiration and early intervention with needle aspiration is usually recommended in order to achieve a complete resolution of the PTX.

We would like to acknowledge some limitations of our systematic review by addressing some points that may provide a more comprehensive understanding of the research landscape in traumatic pneumothorax. Limitations of this review may be attributed to the inclusion criteria and search strings selected for this review. These criteria limit the inclusion of other potentially relevant studies. There could be several reasons why a relatively small number of articles (twelve) specifically addressed conservative management in patients with traumatic pneumothorax. Conducting prospective comparative studies in trauma settings can be challenging due to the urgency and severity of cases and the fact that ethical considerations, patient consent, and logistical issues can restrict the number of high-quality prospective studies in this field. The limited number of studies and the unique characteristics of the subject matter required flexibility in the review process. For this reason, we avoided preliminary registration in the PROSPERO database. This is usually an excellent tool for the prevention of review duplication and bias. However, preliminary review registration could not accommodate the dynamic nature of ongoing reviews on some emerging topics where data are scarce. We aimed to minimize biases by adhering to a transparent and replicable methodology, thus enhancing our review’s reliability and validity. We also tried to systematically search for and screen all relevant articles, extract data using predefined criteria, and synthesize the findings in a structured manner. While recognizing the limitations, which we have acknowledged and discussed thoroughly, we are confident that our review can meet the criteria of a systematic review based on the defined methodology and adherence to PRISMA guidelines. The selected articles for our review encompassed a diverse range of study designs, including observational studies, retrospective analyses, case series, and case reports. This inherent heterogeneity in study methodologies and data collection processes contributed to challenges in conducting direct comparisons and extracting specific management strategies and outcome data from individual centers. The scarcity of controlled prospective trials (only two) and the limited number of high-quality comparative studies addressing our research questions further constrained our ability to extract comprehensive data on management strategies and associated outcomes. In many cases, the available evidence consisted of small sample sizes, varied patient populations, and different clinical contexts. This made it challenging to draw definitive conclusions or conduct meaningful subgroup analyses.

## 6. Conclusions

The modern management of traumatic PTX has shifted toward more conservative approaches, such as smaller catheters or observation [28,29,30]. 

The traditional mantra calling for large-bore chest tubes as first-line approaches to traumatic PTX has been challenged by recent studies demonstrating that pigtail catheters can be equally efficacious alternatives. For these reasons, it is reasonable to adopt a noninvasive approach with observation or needle aspiration for patients in stable conditions under the assumption that trauma patients may benefit from this shift because it reduces their length of stay, complications, and pain.

Our review of conservative management comparing needle aspiration and tube thoracostomy for traumatic pneumothorax highlights the nuanced considerations involved in selecting the most appropriate intervention. The simple needle aspiration procedure appears to be a viable and safe alternative, particularly in non-ventilated or non-positive pressure ventilation patients when hemodynamic conditions remain stable, dyspnea is not present, and there are no other risk factors for underlying chronic lung diseases. To mitigate potential complications associated with tube thoracostomy, a patient-centric approach should be utilized in the management of a traumatic pneumothorax, favoring needle aspiration in select cases as a possible therapeutic option. While acknowledging the existing literature on conservative approaches to traumatic PTX, our case report offers a unique contribution by illustrating the successful resolution of a sizable pneumothorax through needle aspiration.

## 7. Future Directions

There is a paucity of high-grade evidence from clinical trials on recommendations for the conservative treatment of patients with traumatic pneumothoraces. Major limitations of the literature include nonstandardized interventions, a lack of information on the clinical course, and an absence of risk stratification in different clinical settings. There is also insufficient data to recommend conservative therapy for small traumatic pneumothoraces in patients undergoing positive pressure ventilation.

Future medical research on conservative vs. invasive treatment of traumatic pneumothorax should focus on several key areas. This will further advance our understanding and improve patient outcomes. Some potential future research topics include the following:

-Assessing comparative effectiveness. Conducting large-scale randomized controlled trials (RCTs) comparing conservative management (observation, oxygen therapy) with invasive interventions (chest tube insertion, needle decompression) in terms of patient outcomes such as mortality rates, length of hospital stay, complications, and quality of life;-Patient risk assessment and stratification. Identifying specific patient subgroups that may benefit more from conservative management vs. invasive treatment based on factors such as symptomatic or asymptomatic patients, age, comorbidities, mechanism of injury, severity of pneumothorax, presence of other injuries, and comorbidities;-Complications and long-term outcomes. Longitudinal studies assessing long-term outcomes and complications associated with conservative vs. invasive treatment approaches, including rates of recurrence, chronic pain, pulmonary function impairment, and risk of developing pleural adhesions or fibrosis;-Assessing the cost-effectiveness of conservative vs. invasive management strategies based on the utilization of healthcare resources, the direct medical costs, and indirect costs resulting from decreased quality of life and productivity;-Development and implementation of guidelines by updating clinical practice guidelines based on the latest evidence and consensus recommendations to guide healthcare professionals in the management of traumatic pneumothorax, including recommendations for initial assessment, treatment selection, and follow-up care.

## Figures and Tables

**Figure 1 medicina-60-00548-f001:**
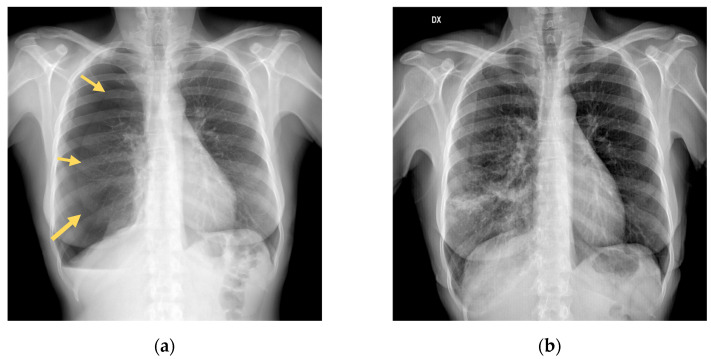
(**a**) Initial chest X-ray shows a right pneumothorax of 55 mm (The arrows mark the right lung border along the pleural line). (**b**) Chest X-ray evaluation, after needle aspiration, shows expansion of the lung parenchyma with a reduction in pneumothorax size.

**Figure 2 medicina-60-00548-f002:**
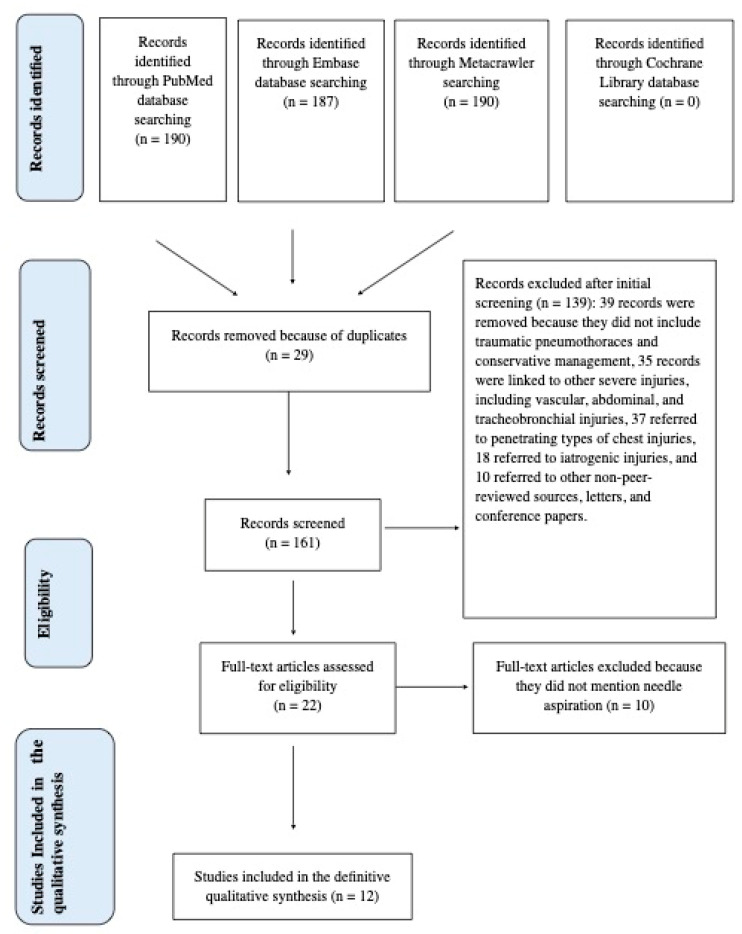
PRISMA flow chart diagram.

**Table 1 medicina-60-00548-t001:** Data on twelve selected papers [13,14,15,16,17,18,19,20,21,22,23,24] on conservative management of PTX, including the first author, the year of publication, the study reference, patient groups, the type of study, and outcomes (LOS, length of stay; RCT, randomized clinical trial; RD, respiratory distress).

Study Reference	Patient Group	Study Type and Level of Evidence	Outcomes	Key Results
Johnson et al., 1996 [13]	29 patients	Retrospective study	Progression to chest drainage	2/29 required chest drain for radiological progression
Banks et al., 2023 [14]	73 patients with pneumothorax size < 30 mm	Retrospective study	LOS with secondary outcomes of pulmonary infection, failed trial of observation, readmission, and mortality	39/73 observation <LOS than thoracostomy group
Partyka et al., 2023 [15]	181 patients with suspected PTX	Retrospective study	Prehospital management: 75 patients out of 181 with traumatic PTX were safely identified and transported without needle decompression to the hospital.	41.4% managed conservatively 58.6% underwent pleural decompression
Anderson et al., 2023 [16]	266 patients with traumatic PTX	Review		90% treated successfully without surgical intervention or subsequent tube drainage
Mattilla et al., 1981 [17]	511 patients with penetrating thoracic injuries	Case series	-	117/511 tube thoracostomy 88/511 needle aspiration
Walker et al., 2018 [18]	602 TARN patients 277/602 treated conservatively	Observational study	Progression to tube drainage intervention: 90% of patients were managed conservatively and did not require tube drainage	Mean ISS 26 252/277 (90%) did not require subsequent thoracic intervention
Ramirez et al., 2012 [19]	31 patients	RCT (Manual aspiration vs Closed tube thoracostomy)	LOS, number of complications	16/31 MA: <LOS, minimal use of analgesia, no need for antibiotic therapy
Kirkpatrick et al., 2013 [20]	90 patients	RCT (Observation vs pleural drainage)	Progression to respiratory distress	No difference in RD
Obeid et al., 1985 [21]	17 patients	Observational study	-	16/17 catether aspiration: no complications, no hospitalization, less cost
Panjwani et al., 2017 [22]	1 patient	Case report	-	Successfully treated with O_2_ administration
Delius et al., 1989 [23]	16 patients	Retrospective study	-	12/16 catheter aspiration (<LOS, less cost) 4/16 progressed to tube thoracostomy
Tran et al., 2021 [24]	-	Review	-	Modern management of traumatic PTX is shifting toward more conservative management practices (smaller catheters or observation)

## Data Availability

Not applicable.

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
