# Peer review of "Successful Needle Aspiration of a Traumatic Pneumothorax: A Case Report and Literature Review"

_medicina, 2024, doi:10.3390/medicina60040548_

Round 1

Reviewer 1 Report (New Reviewer)

Comments and Suggestions for Authors

Thank you for submitting an interesting review article.

I am glad to have a chance to review such a paper. I have the following concerns.

Major comments

1) Even in cases of spontaneous pneumothorax or traumatic pneumothorax, there are cases that can be managed conservatively, which we often encounter clinically and as reviewed by you, there are several reports. Please consider how to show the novelty of this paper.

2) Regarding the discussion section, please do not just state the results of each literature, but compare and integrate them, examine the reasons, and discuss how they relate to the cases you experienced, such as what kind of characteristics they had.

Minor comments

1) Abstract

One of the purposes of this study is to conduct a literature review. In the abstract, please include the results of the literature review and the relevance to this case report.

2) The instructions for authors state the following about review articles: The structure can include an Abstract, Keywords, Introduction, Relevant Sections, Discussion, Conclusions, and Future Directions. Please reconsider the structure if necessary.

3) 3. Materials and Methods
Regarding the literature review, instead of starting abruptly with “3. Materials and Methods”, make a paragraph about the literature review and state its purpose clearly.

4) If you can clearly state the inclusion criteria and exclusion criteria for narrowing down the papers when conducting a literature review, please state them. Also, are patients who use artificial respirators included in the study? How did you exclude such patients from the target study?

5) Technical details of pneumothorax needle aspiration
The paragraph on the technical details of pneumothorax needle aspiration seems unnecessary in this paper.

6) Table 1.

What is the difference between “No. RCT” and “Not RCT” in the study weakness section? Also, I think that the ones with study type as review or case series are not RCTs, but they do not have “No RCT” or “Not RCT” written. Is that necessary?

Thank you for submitting an interesting manuscript.

I am glad to have a chance to review such a paper and wait revised paper.

Author Response

REVIEWER N.1

Major comments

Comment n 1) Even in cases of spontaneous pneumothorax or traumatic pneumothorax, there are cases that can be managed conservatively, which we often encounter clinically and as reviewed by you, there are several reports. Please consider how to show the novelty of this paper.

Response to comment n 1

Thank you for your insightful observation. We acknowledge that conservative management is a common approach in certain cases of traumatic pneumothorax (PTx), particularly when deemed safe by the attending physician. As discussed in our paper, there exists a body of literature detailing successful conservative management of selected patients in different clinical setting.

In our discussion, we emphasized the novelty of our case report by highlighting that it represents the first documented instance in the medical literature where a patient with a sizable traumatic pneumothorax achieved complete resolution following needle aspiration. This distinction underscores the uniqueness of our case and contributes to the evolving understanding of management strategies in traumatic PTx.

To further delineate the novelty of our report, we added some additional considerations in the Discussion section:

The following are some points that highlight the novelty of our case report:

"A few considerations of PTX size can help to highlight that our case is noteworthy for the successful resolution of a large traumatic pneumothorax with an inter-pleural distance measuring 55 mm a substantial difference between this size and the 3.5 cm threshold commonly suggested for conservative management. By demonstrating the efficacy of needle aspiration in such cases, our report challenges existing paradigms. It expands the spectrum of feasible interventions for patients with large traumatic PTX in selected clinical settings with hemodynamic stability without other associated risk factors. Another remarkable aspect of our case is the timing of intervention. Needle aspiration was performed approximately 12 hours post-trauma, suggesting its utility even in a delayed setting. This temporal dimension adds another valuable perspective, particularly in scenarios where immediate intervention may not be feasible or indicated."

And we added in the Conclusion:

“While acknowledging the existing literature on conservative approaches to traumatic PTX, our case report offers a unique contribution by illustrating the successful resolution of a sizable pneumothorax through needle aspiration.”

Comment n 2. Regarding the discussion section, please do not just state the results of each literature, but compare and integrate them, examine the reasons, and discuss how they relate to the cases you experienced, such as what kind of characteristics they had.

Response to comment n 2.

Thank you for your suggestion:

We added the following considerations in Discussion in relation to our case report outlining how the lack of literature can underscore the importance of our case report. We added the following considerations in the Discussion of our case report:

"The main question was whether stable patients with traumatic pneumothoraces could be treated safely and effectively without a chest drain. Additionally, it was also necessary to understand which patients would be the most appropriate for conservative treatment."

"We aimed to contribute in this review to the ongoing discussion about management strategies for a subset of stable patients with large-sized PTX by emphasizing these distinctive features and limitations of the literature where precise indications for conservative management are still unclear. Our case report describing a successful needle aspiration experience in a context of large sized PTX may provide another valuable insight into traumatic conservative therapy in a patient with hemodynamically stable conditions after more than 12 hours from the car accident. A significant contribution to the decision for conservative management was the multidisciplinary discussion with trauma team surgeons, who also recommended needle aspiration as the first line of treatment."

Minor comments

Comment n 1) Abstract. One of the purposes of this study is to conduct a literature review. In the abstract, please include the results of the literature review and the relevance to this case report.

Response to minor comment 1

Thank you for your suggestion; we added the following statements in the Abstract:

“Although recent medical literature supports conservative management of small traumatic PTX guidelines are lacking for hemodynamically stable patients with significant large PTX. The case report documents our successful experience with needle aspiration in such a setting of large traumatic PTX”

 “Our case report offers a novel contribution by illustrating the successful resolution of a sizable pneumothorax through needle aspiration, suggesting that even a large PTX in a hemodynamically stable patient without other risk conditions, can be successfully treated conservatively with simple needle aspiration to avoid tube thoracostomy complications.”

Comment n 2) The instructions for authors state the following about review articles: The structure can include an Abstract, Keywords, Introduction, Relevant Sections, Discussion, Conclusions, and Future Directions. Please reconsider the structure if necessary.

Respsonse to minor comment n 2

Thank you for your suggestion; we added a paragraph of “Future Directions”:

7. “Future Directions:

There is a paucity of high-grade evidence from clinical trials on which recommendations for the conservative tretment of patients with traumatic pneumothoraces. Major limitations of the literature include nonstandardized interventions, lack of information on the clinical course, and absence of risk stratification in different clinical settings. There is also insufficient data to recommend conservative therapy for small traumatic pneumothoraces in patients undergoing positive pressure ventilation.

Future medical research on conservative versus invasive treatment of traumatic pneumothorax should focus on several key areas. This will further advance our understanding and improve patient outcomes. Here are some potential future research topics:

-Assessing comparative effectiveness. Conducting large-scale randomized controlled trials (RCTs) comparing conservative management (observation, oxygen therapy) versus invasive interventions (chest tube insertion, needle decompression) in terms of patient outcomes such as mortality rates, length of hospital stay, complications, and quality of life.

-Patient Risk Assessment and stratification: Identifying specific patient subgroups that may benefit more from conservative management versus invasive treatment based on factors such as symptomatic or asymptomatic patients, age, comorbidities, mechanism of injury, severity of pneumothorax, presence of other injuries, and comorbiditis.

-Complications and Long-Term Outcomes: Longitudinal studies assessing long-term outcomes and complications associated with conservative versus invasive treatment approaches, including rates of recurrence, chronic pain, pulmonary function impairment, and risk of developing pleural adhesions or fibrosis.

-Assessing the cost-effectiveness of conservative versus invasive management strategies, based on the utilization of healthcare resources, the direct medical costs, and indirect costs resulting from decreased quality of life and productivity.

-Development and implementation of guidelines: Updating clinical practice guidelines based on the latest evidence and consensus recommendations to guide healthcare professionals in the management of traumatic pneumothorax, including recommendations for initial assessment, treatment selection, and follow-up care.”

Commnet n 3) Materials and Methods
Regarding the literature review, instead of starting abruptly with “3. Materials and Methods”, make a paragraph about the literature review and state its purpose clearly.

Response to comment n. 3.

Thank you for your suggestion.

We added the following sentences clarifying the purpose of the review prior to Material and Methods

"As a result of our case report, we thought it would be helpful to provide a review of the current evidence related to conservative treatment for traumatic PTX that should not be considered a clinical guidance document. Rather, it tries to provide available evidence about clinical practice in the current medical literature addressing some variation that may exist in the management of traumatic pneumothorax"

Comment n 4) If you can clearly state the inclusion criteria and exclusion criteria for narrowing down the papers when conducting a literature review, please state them. Also, are patients who use artificial respirators included in the study? How did you exclude such patients from the target study?

Response to comment n. 4

Thank you for your comment.

As already reported in the paragraph “Future Directions” there are major limitations in the available literature with a diffuse lack of specific information on the clinical course of patients treated conservatively. The absence of risk stratification in different clinical settings, unavailable data on patients on mechanical ventilation or the percentage of patients on positive pressure ventilation does not allow us to compare and understand the possible benefit of conservative treatment in these specific clinical settings. Only one observational study on a total of 602 patients reported by Walker in 2018 concluded that “From a total of 602 patients with traumatic pneumothorax identified from the database, 277 (46%) were initially managed conservatively without needle decompression, chest tube insertion, or chest surgery. Of the 277 patients managed conservatively, 252 (90%) did not require subsequent intervention, including the majority (56/62, 90%) of patients requiring positive pressure ventilation (PPV). There was no difference in the risk of failure of conservative treatment between ventilated and non-ventilated patients.”

We added the following sentence to further clarify this concept:

The main question was whether stable patients with traumatic pneumothoraces could be treated safely and effectively without a chest drain. Additionally, it was also necessary to understand which patients would be the most appropriate for conservative treatment. A retrospective observational study…

As far as inclusions and exclusion criteria we have included them in the material and methods:

Inclusion criteria: all studies or articles that specifically address the review's research question or topic were considered including randomized controlled trials (RCTs), cohort studies, case-control studies, systematic reviews, or meta-analyses, observational and retrospective studies. Only studies that involve a specific intervention of needle aspiration or conservative treatment following traumatic PTx were considered. The search was conducted across three databases, including Metacrawler, PubMed, and EMBASE from January 1976 to October 2023 employing the following MeSH terms: “Pneumothorax” AND “Truama” AND “Conservative Management”

Exclusion criteria: duplicate publications or multiple reports of the same study, editorials, commentaries, letters, conference abstracts, or other non-peer-reviewed sources as well as articles conducted on animals were excluded. We excluded articles published in languages other than English. Additionally, we excluded all articles that did not include all three MeSH terms. Only peer-reviewed articles were considered.

Comment n 5. Technical details of pneumothorax needle aspiration
The paragraph on the technical details of pneumothorax needle aspiration seems unnecessary in this paper.

Respsonse to minor comment 5.

Thank you for your comment and I agree with your opinion. However, in a previous revision it was specifically requested by one reviewer. Therefore, we hope that shortening its length will be acceptable.

Comment n 6. Table 1. What is the difference between “No. RCT” and “Not RCT” in the study weakness section? Also, I think that the ones with study type as review or case series are not RCTs, but they do not have “No RCT” or “Not RCT” written. Is that necessary?

Respsonse to minor comment 6.

Thank you very much for your observation. We agree. In Table 1, it is evident that all articles are nonrandomized controlled trials. The vertical column at the end of the table has been deleted.

Reviewer 2 Report (Previous Reviewer 2)

Comments and Suggestions for Authors

The initial management of pneumothorax is decided based on the patient’s clinical stability, pneumothorax size, and risks of recurrence. 

The authors' choice of topic is crucial for this reason. In the literature search, the authors selected 12 articles that discussed needle aspiration in the management of traumatic pneumothorax. All the selected articles in PRISMA chart have a significant number of patients (except the case report of Panjwan, which is only a case). So the conclusion is based on an important experience.

The material and method are presented in a clear and comprehensive manner, just like the introduction. The results are based on the selected articles. In my opinion, the discussion can be prolonged, by using more references. Maybe insist on the side effects of single needle aspiration in the traumatic pneumotorax. 

I want to congratulate you for this article, well written and actual. 

Good luck!

I  want to congratu;ated for this ideea

Author Response

REVIEWER N 2

The initial management of pneumothorax is decided based on the patient’s clinical stability, pneumothorax size, and risks of recurrence. 

The authors' choice of topic is crucial for this reason. In the literature search, the authors selected 12 articles that discussed needle aspiration in the management of traumatic pneumothorax. All the selected articles in PRISMA chart have a significant number of patients (except the case report of Panjwan, which is only a case). So the conclusion is based on an important experience.

The material and method are presented in a clear and comprehensive manner, just like the introduction. The results are based on the selected articles. In my opinion, the discussion can be prolonged, by using more references. Maybe insist on the side effects of single needle aspiration in the traumatic pneumotorax. 

I want to congratulate you for this article, well written and actual. 

Good luck!

Response to comments

Thank you very much for your time and insightful comments and positive feedback on our manuscript. It is a pleasure to hear from you that our manuscript does not require any specific changes only a deeper discussion. For this reason, we have added two paragraphs entitled "Future Directions" and "Needle Aspiration Failure".

First paragraph

“7. Future Directions: There is a paucity of high-grade evidence from clinical trials on which recommendations for the conservative tretment of patients with traumatic pneumothoraces. Major limitations of the literature include nonstandardized interventions, lack of information on the clinical course, and absence of risk stratification in different clinical settings. There is also insufficient data to recommend conservative therapy for small traumatic pneumothoraces in patients undergoing positive pressure ventilation.

Future medical research on conservative versus invasive treatment of traumatic pneumothorax should focus on several key areas. This will further advance our understanding and improve patient outcomes. Here are some potential future research topics:

-Assessing comparative effectiveness. Conducting large-scale randomized controlled trials (RCTs) comparing conservative management (observation, oxygen therapy) versus invasive interventions (chest tube insertion, needle decompression) in terms of patient outcomes such as mortality rates, length of hospital stay, complications, and quality of life.

-Patient Risk Assessment and stratification: Identifying specific patient subgroups that may benefit more from conservative management versus invasive treatment based on factors such as symptomatic or asymptomatic patients, age, comorbidities, mechanism of injury, severity of pneumothorax, presence of other injuries, and comorbiditis.

-Complications and Long-Term Outcomes: Longitudinal studies assessing long-term outcomes and complications associated with conservative versus invasive treatment approaches, including rates of recurrence, chronic pain, pulmonary function impairment, and risk of developing pleural adhesions or fibrosis.

-Assessing the cost-effectiveness of conservative versus invasive management strategies, based on the utilization of healthcare resources, the direct medical costs, and indirect costs resulting from decreased quality of life and productivity.

-Development and implementation of guidelines by updating clinical practice guidelines based on the latest evidence and consensus recommendations to guide healthcare professionals in the management of traumatic pneumothorax, including recommendations for initial assessment, treatment selection, and follow-up care.”

Second paragraph

“Needle aspiration failure"

"It is important to consider the possibility of needle aspiration failure in a traumatic pneumothorax, which can involve a number of factors, including inadequate placement of the needle as well as the inability to evacuate air completely from the pleural space due to loculated air pockets. Aspiration failure has been reported more frequently in patients with traumatic PTx with an inter-pleural distance >20 mm at the level of the hilum.  Additionally, if the pneumothorax is large or under tension, needle aspiration alone may not be sufficient to achieve adequate lung re-expansion [26]. In cases where initial needle aspiration fails to fully resolve the pneumothorax, it may be necessary to consider repeat needle aspiration. Some controversies exist on the decision to repeat the procedure in traumatic PTx. The decision is strictly depending on the clinical status of the patient, the size of the pneumothorax, and the expertise of the healthcare provider performing the procedure. It is possible to perform repeated needle aspirations in non-traumatic PTX, such as secondary PTX and iatrogenic PTX. It is possible to perform a second aspiration in patients with traumatic PTx in the absence of respiratory compromise if the PTx persists or recurs. However, when more than 2.5 liters of air are aspirated, the procedure should be considered failed. The time frame from the acute event to the procedure can be an important variable for the success rate of needle aspiration and early intervention with needle aspiration is usually recommended to achieve a complete resolution of the PTx.”

Reviewer 3 Report (New Reviewer)

Comments and Suggestions for Authors

Review of the Manuscript " Successful Needle Aspiration of a Traumatic Pneumothorax: a Case Report and Literature Review

The objective of the study was to present a case of large pneumothorax in a blunt chest trauma patient that resolved with needle aspiration, and to conduct a systematic review summarizing the existing evidence on needle aspiration in traumatic pneumothorax as well as the available evidence supporting conservative management in traumatic PTX. The manuscript is well-written, with clear and concise language that enhances the accessibility of the content.

Having thoroughly examined the content, I would like to pose a few clarifying questions to better understand certain aspects of the study.

·         In the Materials and Methods section, you mention using MeSH terms "Pneumothorax," "Trauma," and "Conservative Management" for your literature review. Could you elaborate on the specific criteria used for including or excluding studies, especially concerning patient populations and types of traumatic pneumothoraces considered?

·         The manuscript discusses a shift towards conservative management for traumatic pneumothoraces. How does the proposed approach align with existing guidelines, such as ATLS, and what aspects of the guidelines support or challenge the adoption of needle aspiration over chest tube insertion?

·         In the Case Presentation section, the patient underwent needle aspiration after multidisciplinary discussions. Could you elaborate on the criteria used to determine the appropriateness of needle aspiration for this specific patient, considering factors like pneumothorax size, stability, and time elapsed since the trauma?

·         Given the potential complications associated with chest tube insertion and needle aspiration, how do you recommend balancing the risks and benefits when choosing between the two interventions? Are there specific patient characteristics or clinical scenarios where one method might be favored over the other?

·         In the Discussion section, you acknowledge limitations such as the limited number of studies addressing conservative management of traumatic pneumothorax. How do you propose mitigating potential bias introduced by the scarcity of data, and are there plans for future research to address these limitations and further validate the findings presented in this manuscript?

Kindly incorporate the responses within the manuscript to augment its overall quality.

Author Response

REVIEWER N 3

The objective of the study was to present a case of large pneumothorax in a blunt chest trauma patient that resolved with needle aspiration, and to conduct a systematic review summarizing the existing evidence on needle aspiration in traumatic pneumothorax as well as the available evidence supporting conservative management in traumatic PTX. The manuscript is well-written, with clear and concise language that enhances the accessibility of the content.

Response

Thank you very much for your time and insightful comments and positive feedback on our manuscript.

-Having thoroughly examined the content, I would like to pose a few clarifying questions to better understand certain aspects of the study.

Comment n 1. In the Materials and Methods section, you mention using MeSH terms "Pneumothorax," "Trauma," and "Conservative Management" for your literature review. Could you elaborate on the specific criteria used for including or excluding studies, especially concerning patient populations and types of traumatic pneumothoraces considered?

Response to comment n 1

Thank you for your observation.

We added in material and methods Inclusion and Exclusion Criteria:

"Inclusion criteria: all studies or articles that specifically address the review's research question or topic were considered including randomized controlled trials (RCTs), cohort studies, case-control studies, systematic reviews, or meta-analyses, observational and retrospective studies. Only studies that involve a specific intervention of needle aspiration or conservative treatment following traumatic PTx were considered. The search was conducted across three databases, including Metacrawler, PubMed, and EMBASE from January 1976 to October 2023 employing the following MeSH terms: “Pneumothorax” AND “Trauma” AND “Conservative Management”.

Exclusion criteria: duplicate publications or multiple reports of the same study, editorials, commentaries, letters, conference abstracts, or other non-peer-reviewed sources as well as articles conducted on animals were excluded. We excluded articles published in languages other than English. Additionally, we excluded all articles that did not include all three MeSH terms. Only peer-reviewed articles were considered."

Comment n 2. The manuscript discusses a shift towards conservative management for traumatic pneumothoraces. How does the proposed approach align with existing guidelines, such as ATLS, and what aspects of the guidelines support or challenge the adoption of needle aspiration over chest tube insertion?

Response to comment n 2

Thank you for your important observation. In Introduction we stated “ In patients with a traumatic pneumothorax, current Advanced Trauma Life Support (ATLS) guidelines recommend chest tube placement, with the caveat that asymptomatic patients with small pneumothoraces who are not ventilated may be managed with observation or aspiration at the discretion of the providing physician. While observation and aspiration are both acceptable options accoding to ATLS guidelines, they are considered at risk of inducing a tension pneumothorax”. To improve discussion about the absence of clear guidelines on conservative management of traumatic PTX we added the following paragraph in Discussion as “Future Directions

”7.Future Directions:

There is a paucity of high-grade evidence from clinical trials on which recommendations for the conservative tretment of patients with traumatic pneumothoraces. Major limitations of the literature include nonstandardized interventions, lack of information on the clinical course, and absence of risk stratification in different clinical settings. There is also insufficient data to recommend conservative therapy for small traumatic pneumothoraces in patients undergoing positive pressure ventilation.

Future medical research on conservative versus invasive treatment of traumatic pneumothorax should focus on several key areas. This will further advance our understanding and improve patient outcomes. Here are some potential future research topics:

-Assessing comparative effectiveness. Conducting large-scale randomized controlled trials (RCTs) comparing conservative management (observation, oxygen therapy) versus invasive interventions (chest tube insertion, needle decompression) in terms of patient outcomes such as mortality rates, length of hospital stay, complications, and quality of life.

-Patient Risk Assessment and stratification: Identifying specific patient subgroups that may benefit more from conservative management versus invasive treatment based on factors such as symptomatic or asymptomatic patients, age, comorbidities, mechanism of injury, severity of pneumothorax, presence of other injuries, and comorbiditis.

-Complications and Long-Term Outcomes: Longitudinal studies assessing long-term outcomes and complications associated with conservative versus invasive treatment approaches, including rates of recurrence, chronic pain, pulmonary function impairment, and risk of developing pleural adhesions or fibrosis.

-Assessing the cost-effectiveness of conservative versus invasive management strategies, based on the utilization of healthcare resources, the direct medical costs, and indirect costs resulting from decreased quality of life and productivity.

-Development and implementation of guidelines by updating clinical practice guidelines based on the latest evidence and consensus recommendations to guide healthcare professionals in the management of traumatic pneumothorax, including recommendations for initial assessment, treatment selection, and follow-up care.

Comment n 3. In the Case Presentation section, the patient underwent needle aspiration after multidisciplinary discussions. Could you elaborate on the criteria used to determine the appropriateness of needle aspiration for this specific patient, considering factors like pneumothorax size, stability, and time elapsed since the trauma?

Response to comment n 3

Thank you for your comment

In this study we recommend hemodinamic stability in eupneic patients; in the case presentation we wrote that:

“The patient's blood pressure was 135/86, her heart rate was 71/minute; she was eupneic with a respiratory rate of 18/min with an oxygen saturation of 96%.”

As far as PTx size ad time frame form the acute event we wrote a comment on the lack of evidence to recommend a threshold risk of time frame and PTx size; 

the following considerations have also been added in future directions:

“There is a paucity of high-grade evidence from clinical trials on which recommendations for the conservative tretment of patients with traumatic pneumothoraces. Major limitations of the literature include nonstandardized interventions, lack of information on the clinical course, and absence of risk stratification in different clinical settings. There is also insufficient data to recommend conservative therapy for small traumatic pneumothoraces in patients undergoing positive pressure ventilation.

Future medical research on conservative versus invasive treatment of traumatic pneumothorax should focus on several key areas. This will further advance our understanding and improve patient outcomes.

Comment n 4. Given the potential complications associated with chest tube insertion and needle aspiration, how do you recommend balancing the risks and benefits when choosing between the two interventions? Are there specific patient characteristics or clinical scenarios where one method might be favored over the other?

Response to comment 4

Thank you for your comment.

During the Discussion, we attempted to address this issue. In order to clarify, we have changed the previous paragraph as follows:

"Our review led us to consider simple needle aspiration as a viable option for our patient, who presented with a significant traumatic pneumothorax (PTX) and remained hemodynamically stable and eupneic despite the car accident occurring 12 hours prior. This decision was reinforced by collaborative discussions with trauma team surgeons, who advocated for a conservative approach with needle aspiration as the initial intervention.

It's essential to acknowledge that our case report doesn't offer direct evidence supporting conservative management of traumatic pneumothoraces. Rather, it suggests that patients deemed suitable for conservative management by their treating physicians exhibit a low likelihood of requiring subsequent interventions."

Comment n 5. In the Discussion section, you acknowledge limitations such as the limited number of studies addressing conservative management of traumatic pneumothorax. How do you propose mitigating potential bias introduced by the scarcity of data, and are there plans for future research to address these limitations and further validate the findings presented in this manuscript? Kindly incorporate the responses within the manuscript to augment its overall quality.

Response to comment 5.

Thank you for your comment.

In response to your question, we added a paragraph about future research under the heading "Future Directions".

"7. Future Directions

There is a paucity of high-grade evidence from clinical trials on which recommendations for the conservative tretment of patients with traumatic pneumothoraces. Major limitations of the literature include nonstandardized interventions, lack of information on the clinical course, and absence of risk stratification in different clinical settings. There is also insufficient data to recommend conservative therapy for small traumatic pneumothoraces in patients undergoing positive pressure ventilation.

Future medical research on conservative versus invasive treatment of traumatic pneumothorax should focus on several key areas. This will further advance our understanding and improve patient outcomes. Here are some potential future research topics:

-Assessing comparative effectiveness. Conducting large-scale randomized controlled trials (RCTs) comparing conservative management (observation, oxygen therapy) versus invasive interventions (chest tube insertion, needle decompression) in terms of patient outcomes such as mortality rates, length of hospital stay, complications, and quality of life.

-Patient Risk Assessment and stratification: Identifying specific patient subgroups that may benefit more from conservative management versus invasive treatment based on factors such as symptomatic or asymptomatic patients, age, comorbidities, mechanism of injury, severity of pneumothorax, presence of other injuries, and comorbiditis.

-Complications and Long-Term Outcomes: Longitudinal studies assessing long-term outcomes and complications associated with conservative versus invasive treatment approaches, including rates of recurrence, chronic pain, pulmonary function impairment, and risk of developing pleural adhesions or fibrosis.

-Assessing the cost-effectiveness of conservative versus invasive management strategies, based on the utilization of healthcare resources, the direct medical costs, and indirect costs resulting from decreased quality of life and productivity.

-Development and implementation of guidelines by updating clinical practice guidelines based on the latest evidence and consensus recommendations to guide healthcare professionals in the management of traumatic pneumothorax, including recommendations for initial assessment, treatment selection, and follow-up care."

Reviewer 4 Report (New Reviewer)

Comments and Suggestions for Authors

Dear Author,

I have read the manuscript "Successful Needle Aspiration of a Traumatic Pneumothorax: a Case Report and Literature Review". The manuscript provides a case presentation of a needle aspiration of a traumatic PTX in a woman. Then, a full literature review was performed.  

The case was well presented with a proper description of PMH,  physical examination,  important clinical findings, diagnostic methods, type of therapeutic intervention, and important follow-up diagnostic results. The literature review was performed according to PRISMA guidelines. 

The discussion section includes a thorough comparison between the authors' findings and existing literature. Additionally, the strengths and limitations of the study were adequately addressed.

Conclusions were also properly stated. 

Author Response

REVIEWER N 4

Dear Author,

I have read the manuscript "Successful Needle Aspiration of a Traumatic Pneumothorax: a Case Report and Literature Review". The manuscript provides a case presentation of a needle aspiration of a traumatic PTX in a woman. Then, a full literature review was performed.  

The case was well presented with a proper description of PMH, physical examination, important clinical findings, diagnostic methods, type of therapeutic intervention, and important follow-up diagnostic results. The literature review was performed according to PRISMA guidelines. 

The discussion section includes a thorough comparison between the authors' findings and existing literature. Additionally, the strengths and limitations of the study were adequately addressed.

Conclusions were also properly stated.

Response to reviewer n 4

My sincere thanks go out to you for taking the time to read our manuscript and for providing us with insightful comments and positive feedback.

We appreciate your interest in and accessibility to our systematic review and case presentation.

Thank you for letting me know that our manuscript does not require any specific changes. By acknowledging the possibility of a noninvasive approach supported in our study and its potential benefits for selected patients, you support the relevance of our message.

Round 2

Reviewer 1 Report (New Reviewer)

Comments and Suggestions for Authors

Thank you for submitting an interesting review article.

I am glad to have a chance to review such a paper. I have the following concerns.

 1.P2 L91 Only a few prospective randomized controlled trials have been performed.

 Please include a few prospective randomized controlled trials as references.

2.  P3 L109 We decided to perform needle aspiration after .
In multidisciplinary discussions, what are the reasons for deciding to perform needle aspiration? For example, how were diagnostic purpose, treatment planning, and risk assessment?”

3. Regarding the literature review, please carefully reconsider your description and, if necessary, rewrite it, taking into account the following points:

(1). Clarify the purpose of the literature review. What was the literature review conducted for? There could be various reasons concerning this case. For example, 1) the choice of treatment for a pneumothorax, 2) differences in outcomes (such as mortality, hospitalization period, additional interventions, costs) between the treatment group and the conservative group for pneumothorax. Please clearly state the purpose of this review. When looking at L120, which mentions “addressing some variation that may exist in the management of traumatic pneumothorax,” it seems to imply that the method of management is the outcome.

(2). Once the purpose is clear, reconsider the appropriate way to write and present the papers. For instance, the content described by Partyka et al. 2023(14) is about a [patient group] of just [181 patients with suspected PTX], with the outcome being [prehospital management], and the key results are how patients were intervened with prehospital. On the other hand, the content described by Walker et al. 2018(17) is about a [Patient group] of 277/602 treated conservatively, with the [outcome] being progression to intervention, and the [key results] being subsequent thoracic intervention. These seem to be quite different from what is being described. Please ensure that the description aligns with the purpose of the literature review.

(3). If the description is done appropriately, there should be no blank spaces in the research’s [outcome].

4. Figure2, PRISAM flow chart diagram

Regarding the PRISMA flow chart diagram, there might be some inaccuracies in the description. Please verify the following points:

1)In the Records screened section, the number of papers after Records removed because of duplication does not seem to match n=29. Instead, this column appears to have 190 records, with 29 removed records. Is that correct?

2)After removing 129 records from the Records screened (n=161) category, wouldn’t the remaining 32 records be the ones assessed for eligibility in the Full-text articles assessed for eligibility section? The mention of 22 records—is that accurate?

5. Please provide the reference for the article that serves as the basis for the statement, L336 “Aspiration failure has been reported more frequently in patients with traumatic PTx with an inter-pleural distance greater than 20mm at the level of the hilum.”

6. Please provide the reference for the article that serves as the basis for the statement, L345 “However, when more than 2.5 liters of air are aspirated, the procedure.”

Thank you for submitting an interesting manuscript.

I am glad to have a chance to review such a paper and wait revised paper.

Author Response

Reviewer 1

Thank you for submitting an interesting review article.

I am glad to have a chance to review such a paper. I have the following concerns.

1.P2 L91 Only a few prospective randomized controlled trials have been performed….

Please include a few prospective randomized controlled trials as references.

Response to question n1

Thank you for your observation. We added references for prospective randomized clinical trilas after the sentence: Only a few prospective randomized controlled trials have been performed comparing conservative and non-conservative management of traumatic pneumothoraces using chest drains (18, 19) 

  1. P3 L109 We decided to perform needle aspiration after ….
    In multidisciplinary discussions, what are the reasons for deciding to perform needle aspiration? For example, how were diagnostic purpose, treatment planning, and risk assessment?”

Response to question n 2

Thank you for your question. In traumatic PTX, we usually follow ATLS guidelines which recommend a chest tube. However, we can provide further insight into the decision for the multidisciplinary discussions that led to this intervention.

Firstly, the patient presented with symptoms indicative of a significant pneumothorax, including chest pain, tenderness, and dyspnoea, along with radiographic evidence confirming its presence. Given the size of the pneumothorax (55 mm), there was concern regarding its potential to progress and compromise respiratory function.

During our multidisciplinary discussions involving trauma surgeons, emergency physicians, and radiologists, several factors were considered:

  1. Diagnostic clarity: Needle aspiration was deemed necessary to confirm the diagnosis definitively and assess the pneumothorax extent. Despite prior imaging, clinical judgment supported the need for real-time confirmation of air volume and its impact on lung function.
  2. Treatment planning: Needle aspiration offered both diagnostic and therapeutic benefits in this case. By relieving the pressure from the pneumothorax, we aimed to alleviate the patient's symptoms and prevent further deterioration, thus potentially avoiding the need for more invasive interventions like chest tube insertion.
  3. Risk assessment: The decision to proceed with needle aspiration involved a careful evaluation of the risks and benefits. While there are inherent risks associated with any invasive procedure, including infection and pneumothorax exacerbation, the potential benefits of prompt intervention outweigh these risks in the context of our patient's clinical presentation and stability.

Furthermore, considering the patient's preference to avoid hospitalization and her immediate return to our Emergency Department seeking care, expedited intervention was deemed prudent to address her symptoms and mitigate the risk of worsening pneumothorax-related complications.

In summary, the decision to perform needle aspiration was guided by the need for diagnostic clarity, treatment planning to alleviate symptoms and prevent further complications, and a comprehensive risk assessment, all of which were thoroughly discussed and evaluated within our multidisciplinary team.

To clarify all these points, we added the following sentences

“As a matter of fact, needle aspiration offered both diagnostic and therapeutic benefits in this case. By relieving the pressure from the pneumothorax, we aimed to alleviate the patient's symptoms and prevent further deterioration, thus potentially avoiding the need for more invasive interventions like chest tube insertion. The decision to proceed with needle aspiration involved a careful evaluation of the risks and benefits. Risks associated with any invasive procedure, including infection and pneumothorax exacerbation were evident; however, the potential benefits of prompt intervention outweighed these risks in the context of our patient's clinical presentation and stability.”

  1. Regarding the literature review, please carefully reconsider your description and, if necessary, rewrite it, taking into account the following points:

(1). Clarify the purpose of the literature review. What was the literature review conducted for? There could be various reasons concerning this case. For example, 1) the choice of treatment for a pneumothorax, 2) differences in outcomes (such as mortality, hospitalization period, additional interventions, costs) between the treatment group and the conservative group for pneumothorax. Please clearly state the purpose of this review. When looking at L120, which mentions “addressing some variation that may exist in the management of traumatic pneumothorax,” it seems to imply that the method of management is the outcome.

(2). Once the purpose is clear, reconsider the appropriate way to write and present the papers. For instance, the content described by Partyka et al. 2023 (14) is about a [patient group] of just [181 patients with suspected PTX], with the outcome being [prehospital management], and the key results are how patients were intervened with prehospital. On the other hand, the content described by Walker et al. 2018(17) is about a [Patient group] of 277/602 treated conservatively, with the [outcome] being progression to intervention, and the [key results] being subsequent thoracic intervention. These seem to be quite different from what is being described. Please ensure that the description aligns with the purpose of the literature review.

(3). If the description is done appropriately, there should be no blank spaces in the research’s [outcome].

Rspsonse to question 3.1

Clarification of the Purpose of the Literature Review: Thank you. We acknowledge the reviewer's point regarding the need for clarity about the nature of our review. As we stated in the Introduction, we aimed to conduct a systematic review summarizing the existing evidence on needle aspiration in traumatic pneumothorax as well as the available evidence supporting conservative management in traumatic PTX. However, as clearly stated in the limitations of our study there was high heterogeneity in the 12 included and available studies: the selected articles for our review encompassed a diverse range of study designs, including observational studies, retrospective analyses, case series, and case reports. This inherent heterogeneity in study methodologies and data collection processes contributed to challenges in conducting direct comparisons and extracting specific management strategies and outcomes data from individual centers. Furthermore, there was also a limited availability of comparative data and despite our efforts to systematically review the existing literature, the scarcity of comparative studies directly comparing different management strategies for traumatic pneumothorax posed significant limitations. Many of the included articles focused on describing single-center experiences or case presentations rather than comparing treatment strategies. As a result, specific data on mortality rates, hospitalization periods, additional interventions required, and associated costs were often lacking or inconsistently reported across studies. For this reason, we have acknowledged constraints of available evidence in relation to the scarcity of controlled prospective trials (only two) and the limited number of high-quality comparative studies that constrained our ability to extract comprehensive data on management strategies and associated outcomes. In many cases, the available evidence consisted of small sample sizes, varied patient populations, and diverse clinical contexts. This made it challenging to draw definitive conclusions or conduct meaningful subgroup analyses.

In our revised manuscript, we transparently acknowledged these limitations associated with the available evidence base. We also acknowledged the challenges encountered in synthesizing data from heterogeneous studies. By openly discussing these limitations, we aimed to provide readers in a paragraph ”Future Directions” the need for future research efforts to address these gaps in the literature.

The primary aim of our literature review was to assess the existing evidence regarding the management of traumatic pneumothorax, specifically focusing on the role of needle aspiration versus conservative treatment. We aimed to identify first any variations in management strategies. When available, the associated outcomes were reported in Table 1 as indicated in the line “Outcome”, only for mortality rates, hospitalization periods (LOS), and additional interventions such as prehospital management. For further clarification of this point we added the following sentence to the paragraph of “limitation of the study”:

"The selected articles for our review encompassed a diverse range of study designs, including observational studies, retrospective analyses, case series, and case reports. This inherent heterogeneity in study methodologies and data collection processes contributed to challenges in conducting direct comparisons and extracting specific management strategies and outcomes data from individual centers. The scarcity of controlled prospective trials (only two) and the limited number of high-quality comparative studies addressing our research questions, further constrained our ability to extract comprehensive data on management strategies and associated outcomes. In many cases, the available evidence consisted of small sample sizes, varied patient populations, and different clinical contexts. This made it challenging to draw definitive conclusions or conduct meaningful subgroup analyses."

Response to question n 3.2

Many thanks for your insightful observation, with which we completely agree. As a matter of fact in the contest described by Partyka the outcome was incompletely described. In Table 1 the sentence about outcome should be clarified as follows:

Prehospital management: 75 patients out of 181 with traumatic PTX were safely identified and transported without needle decompression to the hospital.

We also clarify about Walker's study outcome changing as follows:

Progression to tube drainage intervention: 90% of patients were managed conservatively and didn’t require tube drainage

Respsonse to question n 3. 3

Thank you for your observation; however as explained before in point 3.1 in some studies, specific data on outcome as mortality rates, hospitalization periods, additional interventions required, and associated costs were lacking or inconsistently reported across studies.

  1. Figure2, PRISAM flow chart diagram

Regarding the PRISMA flow chart diagram, there might be some inaccuracies in the description. Please verify the following points:

1)In the Records screened section, the number of papers after Records removed because of duplication does not seem to match n=29. Instead, this column appears to have 190 records, with 29 removed records. Is that correct?

Response to question 4.1

The following is an explanation of the PRISMA flow chart: The number of records appearing on PubMed and Metacrawler is 190, 3 more than Embase's 187 records. As a result, we considered 190 articles of which 29 were duplicates of previous articles from the same center or authors: 190 - 29 = 161 (records screened).

2)After removing 129 records from the Records screened (n=161) category, wouldn’t the remaining 32 records be the ones assessed for eligibility in the Full-text articles assessed for eligibility section? The mention of 22 records—is that accurate?

Response to question n. 4.2.   

Many thanks for your correct observation; indeed, there were other 10 records not mentioned in the PRISMA flow chart that were removed because they matched “exclusion criteria” (letters, conference abstracts, or other non-peer-reviewed sources). We added the excluded 10 records to the PRISMA flow chart. In conclusion 139 records were excluded after initial screening: of them 39 records were removed because not including “traumatic pneumothorax” AND “conservative management”, 35 records were linked to other severe injuries, including vascular, abdominal, and tracheobronchial, 37 were referred to penetrating types of chest injuries, 18 articles were referred to iatrogenic injuries and 10 were removed because they entered the exclusion criteria (other non-peer-reviewed sources, letters and conference papers). 39+35+37+18+10 = 139 (records excluded). At this point 22 records remained but 10 of them didn’t mention needle aspiration and were removed. This left 12 articles that we selected and reported in Table 1 that were edited for that number.

  1. Please provide the reference for the article that serves as the basis for the statement, L336 “Aspiration failure has been reported more frequently in patients with traumatic PTx with an inter-pleural distance greater than 20mm at the level of the hilum.”

Response to question n 5.

Thank you: we have added the reference number (26). The article was alrady reported in the references.

  1. Please provide the reference for the article that serves as the basis for the statement, L345 “However, when more than 2.5 liters of air are aspirated, the procedure….”

Thank you for your suggestion. A new reference was added, the n 12

Chan SS. The role of simple aspiration in the management of primary spontaneous pneumothorax. J Emerg Med. 2008, 34,131-138.

All other reference numbers were changed in the text and in the reference according to the new one.

Thank you for submitting an interesting manuscript.

I am glad to have a chance to review such a paper and wait revised paper.

Response: Thank you very much for your time and insightful comments; your interest in our systematic review and case presentation is greatly appreciated.

This manuscript is a resubmission of an earlier submission. The following is a list of the peer review reports and author responses from that submission.

Round 1

Reviewer 1 Report

Comments and Suggestions for Authors

The manuscript named <Successful Needle Aspiration of a Traumatic Pneumothorax: a  Case Report and Literature Review> by Bettoni J et al is a systematic review summarising the existing evidence on needle aspiration in traumatic pneumothorax as well as a case presentation of large pneumothorax in a chest trauma patient that resolved with needle aspiration. The main idea of the manuscript is for that reason there are no specific indications for conservative management in hemodynamically stable patients with significant traumatic PTXs after more than 12 hours from the car accident. The authors conclude that a non-invasive approach is beneficial for the patient, taking into account the pain, length of hospitalization and complications, and they argue the position based on a review of the literature and experience. The authors have done an interesting analysis of a complicated clinical case in a simple and accessible way.

Author Response

Response to Reviewer 1

Thank you very much for your time and insightful comments and positive feedback on our manuscript.

Your interest and accessibility in our systematic review and case presentation is greatly appreciated.

It is a pleasure to hear from you that our manuscript does not require any specific changes. Your acknowledgment of the possibility of a non-invasive approach supported in our study and its potential benefits for selected patients endorses the relevance of our message.

Reviewer 2 Report

Comments and Suggestions for Authors

Dear authors,

I came upon a really interesting article. Needle aspiration may be a safe, less invasive, and effective initial treatment option in well-selected patients with pneumothorax. No differences in mortality were observed in the literature between needle aspiration and intercostal tube drainage in pneumothorax.

Came back to your article:

You don't have the second figure. Only 1 and 3?

Figure 3:  remake it, please

Put the number of references for the article included in table 1

More than 75% of references are to old; can you refresh a little? There are more than 5,000 articles on PubMed on this topic.

With a little effort, I think you can significantly improve this article, to meet the rigors imposed by the journal chosen for publication.

Good luck!

Author Response

Dear authors,

I came upon a really interesting article. Needle aspiration may be a safe, less invasive, and effective initial treatment option in well-selected patients with pneumothorax. No differences in mortality were observed in the literature between needle aspiration and intercostal tube drainage in pneumothorax.

Came back to your article:

You don't have the second figure. Only 1 and 3?

Response:

Thanks for your observation. Actually we have only two figures Figures 1 (a and b) and Figure 2. The PRISMA flow chart is Figure 2 and it was incorrectly referred to as Figure 3; the error has been corrected.

Figure 3:  remake it, please

Response:

Thanks. Most likely, the pdf format was displayed erratically in the word document. We have revised PRISMA flow chart (Figure 2)

Put the number of references for the article included in table 1

Response:

Thanks for your suggestion. We put the reference number for each cited paper in Table 1

More than 75% of references are to old; can you refresh a little? There are more than 5,000 articles on PubMed on this topic.

Response:

We appreciate your observation, however although the number of articles about pneumothorax is almost uncountable when you combine traumatic pneumothorax “AND” conservative management, the available articles are strictly limited to those we have cited. Even so, we have added 3 very recent articles on managing traumatic pneumothoraxes; however, only one of them is partly related to conservative management.

With a little effort, I think you can significantly improve this article, to meet the rigors imposed by the journal chosen for publication. Good luck!

Response:

Thank you very much for your time and insightful comments.

Reviewer 3 Report

Comments and Suggestions for Authors

I reviewed the manuscript about needle aspiration of a traumatic pneumothorax. A traumatic pneumothorax is a life threatening condition. Although the case is clearly described, single cases have little scientific value. It is better do describe series of cases or cohorts. Authors added a literature review, which was prepared correctly but is too short. The PRISMA flow chart diagram was included. However there is no clear conclusion about recommendation of treatment of traumatic pneumothorax : needle aspiration or tube thoracostomy. Therefore I regret to recommend rejecting the manuscript.

Author Response

Reviewer n 3

I reviewed the manuscript about needle aspiration of a traumatic pneumothorax. A traumatic pneumothorax is a life threatening condition. 1.Although the case is clearly described, single cases have little scientific value. It is better do describe series of cases or cohorts. 2. Authors added a literature review, which was prepared correctly but is too short. The PRISMA flow chart diagram was included. 3. However there is no clear conclusion about recommendation of treatment of traumatic pneumothorax: needle aspiration or tube thoracostomy. Therefore, I regret to recommend rejecting the manuscript.

Response:

  1. We acknowledge the reviewer's concern regarding the scientific value of single case reports. However, we believe our case report offers unique insights into traumatic pneumothorax management. Specifically, our report outlines the successful drainage of a large traumatic pneumothorax through needle aspiration, a simple and non-invasive procedure that circumvents tube thoracostomy.
  1. To address the brevity of the case report, we have expanded our manuscript to include in Material and Methods a detailed description of the “needle aspiration procedure”. This addition provides readers with a comprehensive understanding of the technique employed in our case, thereby enhancing our report's scientific rigor.
  2. Regarding the conclusion, we appreciate the reviewer's feedback and have revisited our statements to ensure clarity. While we understand the limitations of drawing definitive conclusions from a single case report, we have cautiously articulated our findings. We highlight needle aspiration as a potential alternative to tube thoracostomy, particularly in select patient populations. By emphasizing its feasibility and safety in specific clinical contexts, we aim to stimulate further investigation and consideration of this therapeutic option.

In light of these revisions and clarifications, we respectfully request to reconsider your recommendation to reject the manuscript. We hope that our contributions can merit consideration in the journal, and we remain open to further suggestions.

Reviewer 4 Report

Comments and Suggestions for Authors

Dear Authors,

thank you very much for the opportunity to review the article entitled Successful Needle Aspiration of a Traumatic Pneumothorax: a Case Report and Literature Review. It raises important issues related to a clinically common problem in emergency medicine practice. The manuscript is clear, relevant to the field, and presented in a well-structured way. The article is written in a clear and concise manner, but at the same time sufficient to discuss the strictly focused issue.

A presentation of a case of large pneumothorax (PTX) in a blunt chest trauma patient resolved with needle aspiration and conducting a literature review summarizing the existing evidence on needle aspiration in traumatic pneumothorax as well as the available evidence supporting conservative management in traumatic PTX is an interesting and necessary topic, both scientifically and clinically.

After a thorough analysis of the manuscript, several questions and doubts arise. To improve the quality of the article, it would be worth taking them into account:

MAJOR revision

-    - TITLE of the article and MATERIAL AND METHODS: the title generally states: "Literature review". Line 84 of the Introduction talks specifically about the "systematic review" type. With regard to the Methodology presented in the manuscript, please indicate what type of literature review the authors intended and whether it was intended to be a systematic review and not, for example, a narrative review.

-      - Additionally, in the MATERIAL AND METHODS chapter: were there any other exclusion criteria apart from the mentioned methods of selecting publications for the review? Table 1 presents very different types of publications, including Case report, Case series and 2 Reviews.

MINOR revision

- INTRODUCTION - to complete the introduction to emergency therapeutic options, it is worth adding brief information about finger thoracostomy.

- There are figures in the manuscript marked as Figure 1 and Figure 3. Should there be another figure (Fig 2.) or is it a minor editor's error when numbering?

- Figure 3. is illegible due to overlapping windows. I believe this is a minor editorial issue and a technical error in compiling the figure to the PDF version.

Thank you again for the opportunity to review an article on, after all, these very important issues. I sincerely hope that this review will allow you to draw constructive conclusions that will prove useful.

Author Response

Reviewer n 4

Dear Authors,

thank you very much for the opportunity to review the article entitled Successful Needle Aspiration of a Traumatic Pneumothorax: a Case Report and Literature Review. It raises important issues related to a clinically common problem in emergency medicine practice. The manuscript is clear, relevant to the field, and presented in a well-structured way. The article is written in a clear and concise manner, but at the same time sufficient to discuss the strictly focused issue.

A presentation of a case of large pneumothorax (PTX) in a blunt chest trauma patient resolved with needle aspiration and conducting a literature review summarizing the existing evidence on needle aspiration in traumatic pneumothorax as well as the available evidence supporting conservative management in traumatic PTX is an interesting and necessary topic, both scientifically and clinically.

After a thorough analysis of the manuscript, several questions and doubts arise. To improve the quality of the article, it would be worth taking them into account:

MAJOR revision

- TITLE of the article and MATERIAL AND METHODS: the title generally states: "Literature review". Line 84 of the Introduction talks specifically about the "systematic review" type. With regard to the Methodology presented in the manuscript, please indicate what type of literature review the authors intended and whether it was intended to be a systematic review and not, for example, a narrative review. Additionally, in the MATERIAL AND METHODS chapter: were there any other exclusion criteria apart from the mentioned methods of selecting publications for the review? Table 1 presents very different types of publications, including Case report, Case series and 2 Reviews.

Response:

Thank you for your valuable feedback and for bringing up the question regarding the nature of our literature review.

In our paper we employed a systematic literature review methodology according to PRISMA guidelines. In Materials and Methods, we outlined our search strategy, including the specific search engines we employed. We also outlined the Boolean operators "AND" applied and the MeSH terms utilized. Our intention was to conduct a comprehensive and structured synthesis of the available literature on the topic.

We acknowledge that in the introduction, we referenced the term "systematic review". However, upon closer examination of our methodology section, it becomes quite clear that our approach aligns with the rigorous standards of a systematic review rather than a narrative review.

We added the following sentences for further clarification:

“We aimed to minimize biases by adhering to a transparent and replicable methodology, thus enhancing our review's reliability and validity. We also tried to systematically search and screen all relevant articles, extracted data using predefined criteria synthesizing the findings in a structured manner. While recognizing the limitations, which we have acknowledged and discussed thoroughly, we are confident that our review can meet the criteria of a systematic review. This is based on the defined methodology and adherence to PRISMA guidelines.”

We appreciate your attention to detail and are grateful for the opportunity to clarify the nature of our review. Please let us know if you need further clarification or information.

MINOR revision

- INTRODUCTION - to complete the introduction to emergency therapeutic options, it is worth adding brief information about finger thoracostomy.

Response:

Thank you for your valuable suggestion. According to your comment in the Introduction we added the following statement “In ventilated trauma patients in the pre-hospital setting with impending tension PTX, bilateral finger thoracostomy starting on the side with the suspected tension PTX is a valuable temporary measure followed by definitive tube thoracostomy when the patient arrives in the trauma center for stabilization.”

- There are figures in the manuscript marked as Figure 1 and Figure 3. Should there be another figure (Fig 2.) or is it a minor editor's error when numbering?

Response:

Thanks for your observation. Figures 1 (a and b) and Figure 2. The PRISMA flow chart is Figure 2 and was incorrectly referred to as Figure 3; the error has been corrected.

- Figure 3. is illegible due to overlapping windows. I believe this is a minor editorial issue and a technical error in compiling the figure to the PDF version.

Response:

Thanks. Most likely, the pdf format was displayed erratically in the word document. We have revised the flow chart figure.

Thank you again for the opportunity to review an article on, after all, these very important issues. I sincerely hope that this review will allow you to draw constructive conclusions that will prove useful.

Response:

We appreciate all the insightful comments and positive feedback you provided on our manuscript, and we thank you very much for your time.

Reviewer 5 Report

Comments and Suggestions for Authors

The description of the patient with PTX can include some more data, such as pain, respiratory rate/ tachypnea, dyspnea and blood pressure.

Fig. 3 has technical issues, some statements are not readable in the pdf file.

Only 4/25 references are from the last five years (from 2020 to 2024). Please revise. 

'GRADE approach' has been devised to rate the certainty of evidence in systematic reviews and other evidence syntheses. Although this article is not a systematic review, this approach could be used to provide a more in-depth data on the included articles. 

The data abstracted from the literature should be analyzed focusing on comparisons of therapeutic procedures in PTX. Case reports and review articles will have little value in this regard. 

Author Response

Reviewer n 5

The description of the patient with PTX can include some more data, such as pain, respiratory rate/ tachypnea, dyspnea and blood pressure.

Response:

Thank you for your observation. We added the missing data in the following sentences:She complained of pain in her right chest (Visual Analog Scale score 6).”  The patient's blood pressure was 135/86, her heart rate was 71/minute; she was eupneic with a respiratory rate of 18/min with an oxygen saturation of 96%.

Fig. 3 has technical issues, some statements are not readable in the pdf file.

Response:

Thanks. Most likely, the pdf format was displayed erratically in the word document. We have revised the PRISMA figure.

Only 4/25 references are from the last five years (from 2020 to 2024). Please revise. 

Response:

We appreciate your observation. The number of articles about pneumothorax is almost uncountable; however, when you combine traumatic pneumothorax “AND” conservative management, the articles are strictly limited to those we have cited. Even so, we have added 3 very recent articles on managing traumatic pneumothoraxes, only one of which focuses on conservative treatment.

'GRADE approach' has been devised to rate the certainty of evidence in systematic reviews and other evidence syntheses. Although this article is not a systematic review, this approach could be used to provide a more in-depth data on the included articles. 

The data abstracted from the literature should be analyzed focusing on comparisons of therapeutic procedures in PTX. Case reports and review articles will have little value in this regard. 

Response:

We appreciate your observation. We acknowledge that in the introduction, we referenced the term "systematic review". However, upon closer examination of our methodology section, it becomes quite clear that our approach aligns with the standards of a systematic review following PRISMA guidelines rather than a narrative review.

We added the following sentences in Discussion for further clarification:

“We aimed to minimize biases by adhering to a transparent and replicable methodology, thus enhancing our review's reliability and validity. We also tried to systematically search and screen all relevant articles, extracted data using predefined criteria synthesizing the findings in a structured manner. While recognizing the limitations, which we have acknowledged and discussed thoroughly, we are confident that our review can meet the criteria of a systematic review. This is based on the defined methodology and adherence to PRISMA guidelines.”

In relation to your comment about “Grade approach” which defines the quality of a body evidence as the extent to which one can be confident that an estimate of effect or association is close to the quantity of a specific interest, we would like to stress that we have described in a long and clear paragraph the limitation of the study in the contest of the conservative management of pneumothorax. We would like to stress again these limitations have been described in the Discussion section by addressing some points which can provide a comprehensive understanding of the research landscape in traumatic pneumothorax. We clearly stated that limitations of this review may be attributed to the inclusion criteria and specific search strings selected for this review. These criteria limit the inclusion of other potentially relevant research. There could be several reasons why a relatively small number of articles (twelve) specifically address conservative management in patients with traumatic pneumothorax. Difficulties conducting prospective comparative studies in trauma settings can be challenging due to the urgency and severity of cases since ethical considerations, patient consent, and logistical issues may restrict the number of high-quality prospective studies in this field. Due to a limited literature base and unique characteristics of the subject matter, the review process had to be flexible.

We appreciate your attention to detail and are grateful for the opportunity to clarify the nature of our review. Please let us know if you need further clarification or information and we remain open to further suggestions.